# Does Pool Performance of Elite Triathletes Predict Open-Water Performance?

**DOI:** 10.3390/jfmk8040165

**Published:** 2023-12-06

**Authors:** Sergio Sellés-Pérez, Roberto Cejuela, José Fernández-Sáez, Héctor Arévalo-Chico

**Affiliations:** 1Physical Education and Sports, University of Alicante, 03690 Alicante, Spain; sergio.selles@ua.es (S.S.-P.); hector.arevalochico@ua.es (H.A.-C.); 2Unitat de Suport a la Recerca Terres de l’Ebre, Fundació Institut, Universitari per a la Recerca a l’atenció Primària de Salut Jordi Gol i Gurina (IDIAPGol), 43500 Tortosa, Spain; jfernandez@idiapjgol.info; 3Facultat de Enfermería, Campus Terres de l’Ebre, Universitat Rovira i Virgili, 43500 Tortosa, Spain

**Keywords:** swim, triathlon, test, lactate

## Abstract

The capacity of laboratory tests to predict competition performance has been broadly researched across several endurance sports. The aim of the present study was to analyse how pool swimming performance can predict the result of the swimming segment in triathlon competitions and compare predictability differences based on competition level and distance. Eighteen male triathletes participated in the study. Three were ranked world-class, ten elite/international level, and five highly trained/national level. A total of sixty-one graded multi-stage swimming tests were conducted. Blood lactate was measured to calculate the following hypothetical predictor variables: speed at lactate threshold 1 (LT1), speed at lactate threshold 2 (LT2), and speed in the last repetition of the test (S_L200_). The following data were collected for a total of 75 races: time in the swimming leg (TSL); position after the swimming leg (PSL); time difference with the first triathlete after the swimming leg (DFT); and final race position. The race levels were divided according to participant levels as follows: world series (WS) (*n* = 22); World Cup (WC) (*n* = 22); Continental Cup (CC) (*n* = 19); national championship (N) (*n* = 5); and local race (L) (*n* = 5). Based on distance, they were divided into Olympic distance (OD) (*n* = 37) and sprint distance (SD) (*n* = 38). A moderate to strong positive association was found between LT1, LT2, S_L200_ and PSL and TSl at all race levels except for the SD CC, SD WC, and OD CC races, where no or weak-to-moderate correlations were found. The present study demonstrated that performance measured in a graded multi-stage pool lactate test can predict performance in a triathlon swimming segment. This finding is highly useful for coaches as it can help them to obtain a reliable measure of the triathlete’s specific capabilities in the swimming leg.

## 1. Introduction

A triathlon is a multidisciplinary endurance sport where swimming, cycling, and running succeed each other without stopping the chronometer [1]. Triathlon includes a wide range of distances and modalities but competitions can be generally divided into short-distance triathlons such as sprint (SD) or Olympic (OD) and long-distance triathlons, e.g., Ironman. OD (1.5 km swimming, 40 km cycling, and 10 km running) is performed at the Olympic Games together with mixed relay. SD (750 m swimming, 20 km cycling, and 5 km running) is another common distance in international and national-level races. The World Triathlon Series (WS) are the highest international level competitions, followed by the World Cup (WC) and the Continental Cups (CC). The points athletes obtain in these competitions define their international ranking [2]. To be able to participate in the WS, triathletes must accumulate points in lower-level races such as the WC or CC [3]. Moreover, some national federations organise their own national elite championships (N) [3]. Added to these official competitions, it is common for some elite triathletes to participate in local races (L) as part of their training. Both elite athletes and age-group (recreational) triathletes compete together in these races [4].

A triathlon event begins with an open-water swim (lakes or seas). In an OD triathlon, this swim accounts for 10% to 15% of the race [5]. Despite a low race time compared to cycling or running, performance in this segment is very important in short-distance triathlons, because drafting on a bike is commonly allowed. In fact, the overall finishing position of elite triathletes in the OD triathlon is significantly correlated with average swimming velocity and the position after the swim stage [6].

Although a large part of triathletes’ training takes place in the pool, open-water swimming presents a series of particularities such as a mass start, drafting, or frontal breathing—aimed at self-orienting and not swimming more metres than necessary during the segment [7]. For example, regarding drafting, it has been observed how swimming behind another triathlete can reduce oxygen uptake by 10 to 25% [8]. Lateral drafting also reduces energy expenditure and muscle fatigue, although less than behind drafting [9]. It has been suggested that the optimal drafting distance is situating one’s fingertips between 0 m to 0.50 m behind another swimmer’s toes [10]. Another characteristic of triathlon swimming is the use of the wetsuit, which is allowed in elite competitions when the water temperature is below 20 °C [1]. Using a wetsuit is known to have a positive effect on swimming speed [11]. Moreover, a recent investigation observed how using the wetsuit leads to less energy expenditure and a decrease in heart rate, as well as a lower perception of effort compared to when swimming at the same speed in a swimsuit. Differences have also been detected regarding technical and biomechanical parameters, such as a lower number of foot movements, strokes and breaths when using the wetsuit [12].

Owing to the combinations of external factors described above that can affect the triathlon swimming segment, it is difficult to predict competitive performance through laboratory tests. The predictive capacity of laboratory tests has been widely studied in sports science research, particularly in endurance sports. In the case of triathlon, some authors have demonstrated this predictive capacity using various methods and at various distances [13,14,15]. One of the most widespread methods is lactate profiling. Lactate accumulates during high-intensity exercise because lactic acid production levels surpass removal levels. Therefore, by studying the dynamics of blood lactate levels in relation to different speeds, it is possible to obtain an indication of the levels of demand of these intensities for the athlete [16]. It is thus possible to establish certain intensity levels at which lactate is in its maximum steady state, providing highly revealing data of the athlete’s aerobic capacity [17]. For example, a correlation was observed between Olympic triathlon performance and cycling and running power developed at 2–4 mmol [18]. Another common method is to conduct time trials to study their correlation with competition performance. However, the 25 m or 400 m time did not correlate with the outcome of a national championship for elite triathletes in OD. Cuba-Dorado et al. found correlations, albeit very weak ones, between the 400 and 1000 m swimming tests of the Spanish Triathlon Federation talent detection programme and the swimming leg in national competitions [19].

Nevertheless, to our knowledge, no study has been conducted with national-and international-level triathletes on the results of a lactate incremental test regarding triathlon open-water performance. A possible explanation is that researchers have historically faced a series of limitations regarding high-level athlete data collection such as a limited sample size, difficulties in coordinating competition calendars with measurement-taking, data quality, and applicability [19].

Therefore, the study aim was to correlate elite triathlete performance in the pool tests with the athletes’ results in different competition levels (L, N, CC, WC and WTS) and distances (SD and OD). In this way, we sought to determine whether pool test performance could predict competition performance, and the types of triathlons in which the correlation was stronger. Another study objective was to determine whether the wetsuit benefitted triathletes in the pool tests differently, according to their levels. This information could be highly valuable for coaches and athletes to design their training plans based on triathlete capabilities.

Our initial hypotheses were as follows: (1) performance variables measured during pool tests predict competition performance; (2) pool performance variables have a greater performance prediction power in the case of lower level competition races; (3) triathletes presenting a weaker performance in the pool tests benefit more from using wetsuits during competitions.

## 2. Materials and Methods

### 2.1. Participants

A total of eighteen male triathletes participated in the study. Three triathletes were ranked world-class level, ten elite/international level, and five highly trained/national level according to McKay’s framework for research in sport science [20]. Participants included three triathletes with at least one podium finish in WS, five triathletes with at least one podium finish in WC, and four national champions. The triathletes trained in the same squad under the supervision of the same certified coach. All participants were used to open-water swimming training, with approximately 25% of their swimming training sessions conducted in open-water conditions.

Their mean and standard deviation (ds) age, height, and weight were 22.6 (3.4) years, 175.2 (7.8) cm, and 64.2 (4.8) kg, respectively. All participants had over 5 years of experience in triathlon training and competing, the mean (ds) number of years of experience being 6.8 (4.2) years. All participants underwent a medical examination at the beginning of the season to check that they were prepared for high-intensity exercise. The participants gave their prior consent to their data being used in this study. The procedures implemented in this study were approved by the Ethics Committee (Expedient UA 2022-09-29_1) of the University of Alicante. The whole data collection process followed the guidelines of the Declaration of Helsinki.

### 2.2. Study Design

This study was executed in two stages: the swimming performance test and the subsequent analysis of the race data. The testing phase focused on conducting a graded multistage test with lactate analysis at the pool and a 200 m time trial (t200). A total of sixty-one graded multi-stage swimming tests and time trials were conducted. These measurements were timed to be as close as possible to the competition and were taken at the beginning of each competitive period, when the majority of competitions took place. Each competitive period had a maximum duration of 4 microcycles, each microcycle lasting one week. Figure 1 shows an example of the study design for one participant.

### 2.3. Blood Lactate Concentration Data Analyses

Swimming tests were performed as a graded multistage test consisting of seven 200 m repetitions in a 5 min cycle, graded from easy to maximal [21]. Target times were individualised based on each triathlete’s personal best 200 m times calculated on the day before each test. The speed of the seventh and last test repetition (S_L200_) was calculated and served as a measure of maximal training performance [22]. Blood lactate concentrations were measured to calculate the speed at lactate threshold 1 (LT1) and speed at lactate threshold 2 (LT2). Blood samples taken from the earlobe were examined using a portable lactate analyser (Lactate Pro, Arkray Inc., Amstelveen, The Netherlands). The standards for identifying thresholds were set as follows: a 0.5 mMol/L rise in blood lactate for LT1 and a rise of over 1.0 mMol/L for LT2 [16,17]. All tests were conducted in the same 50 m heated pool at a temperature of 26 °C.

### 2.4. Race Data

A minimum of 29 races was necessary to obtain a correlation coefficient of at least 0.500 with a significance level of 0.05 and a statistical power of 80%. A total of 75 races were analysed. The races were divided according to competitive level into WS (*n* = 22), WC (*n* = 22), CC (*n* = 19), N (*n* = 5), and L (*n* = 5). Based on distance, they were divided into OD (*n* = 37) and SD (*n* = 38). Competitions that were conducted with wetsuits and without wetsuits were also noted. Table 1 shows all the races, wetsuit and no wetsuit competitions, and the number of participants that were analysed.

The following data were collected: time in the swimming leg (TSL), position after the swimming leg (PSL), time difference with the first triathlete after the swimming leg (DFT), and final race position (FRP). All the data are available on the official website of the international triathlon federation [23] and on the official website of the Spanish triathlon federation [18].

### 2.5. Statistical Analysis

A correlation analysis using Pearson and Spearman correlation coefficients was conducted to verify competition performance predictability based on the variables measured in the swimming tests. To determine whether the use of wetsuits could benefit triathletes to a greater or lesser extent based on their pool test performance, the rate of intrasubject variation (RV) between TSL with wetsuits and TSL without wetsuits was calculated. A correlation analysis was also conducted using Spearman correlation coefficients between RV and LT1, LT2, and S_L200_. The Rhea criterion was used to determine correlation strength [24]. The statistical software Statistical Package for Social Sciences (SPSS) 22.0 (SPSS Inc., Chicago, IL, USA) was used to analyse the data and calculate the statistical power. For all analyses, significance was accepted at *p* < 0.05.

## 3. Results

Table 2 shows the average results of the pool performance tests based on the participants’ competitive level. No significant differences were found between the World Class Level group and the Elite Level group in any swimming performance variable. No statistically significant differences were found either between the World Class Level group and the National Level group in LT1, LT2, S_L200_, and the 200 m swim test, nor between the Elite Level group and the National Level group.

Table 3 shows the Pearson and Spearman correlation coefficient between the test variables and DFT in the different race levels. Moderate to strong negative correlations were found with LT1, LT2, and S_L200_, while positive correlations were observed with t200 across all race levels except for SD CC, SD WC, and OD CC races, where no or weak-to-moderate correlations were found. In most of the race levels, the strongest correlations were found with S_L200_, followed by LT2 and LT1. The lowest correlation was with t200. Regarding race distance, no correlation strength differences were found between SD and OD, except in the L races where the correlations in OD were stronger than in SD and in WC, where significant correlations were found in OD but not in SD.

Table 4 also shows the Pearson and Spearman correlation coefficient between the test variables and PSL across the various race levels. Moderate to high negative correlations were found with LT1, LT2, and S_L200_, while positive correlations were observed with t200 in all race levels. No or weak-to-moderate correlations were found, however, in the SD WC, and OD CC races. In most race levels, the strongest correlations were found with S_L200_, LT2 and t200. The lowest correlation was with LT1. Analysing the race distance variable, correlation strength differences were found between SD and OD in L races, where the correlations in OD were stronger than in SD; in CC races, where significant correlations were found in SD but not in OD; in WC, where significant correlations were found in SD but not in OD; and in WS, where the correlations in SD were slightly stronger than in OD.

Finally, Table 5 shows the Pearson and Spearman correlation coefficient between the test variables and FRP across the various race levels. Moderate to high correlations were only found in the SD WS, OD N and OD WS races.

The RV in the TSL variable in SD triathlons measured in competitions without a wetsuit and with a wetsuit showed a statistically significant negative correlation with S_L200_ (q = −0.622; *p* < 0.05). No other correlations were found with any other variable and nor with OD races.

## 4. Discussion

The aim of this study was to analyse the predictive power of the results of a lactate profiling test conducted in a pool regarding the swimming performance of world, international, and national level triathletes in triathlons.

The most significant finding of this study was that the variables measured in the pool test were significantly correlated with the subsequent performance in the swimming leg across most competitive levels.

The triathlon swimming leg has several intrinsic characteristics that differentiate it from pool swimming, such as mass starts, drafting and frontal breathing [7]. However, in most competitive levels, high LT1, LT2, S_L200_ values and low t200 values coincide with low DFT and PSL values. These results are similar to those found for elite-level swimmers [25].

### 4.1. Correlations between Test Variables and DFT

In triathlon events, most of the athlete’s expended energy comes from aerobic metabolism [26], so triathletes who are better adapted to this type of effort are logically more likely to finish in better positions. Therefore, swimming performance in triathletes is mainly determined by energy demands and interactions with biomechanical variables [27]. Puccinelli et al. demonstrated that maximal aerobic speed was the best predictor of performance in the swimming segment in the case of amateur triathletes [13]. Although this variable was not measured in our study, high S_L200_ and LT2 values similarly correlated with DFT. Short-distance triathlon competitions are also characterised by high-intensity intermittent phases, so developing both aerobic and anaerobic capacity allows athletes to withstand these constant intensity changes [7].

The t200 variable also showed statistically significant correlations with DFT. This finding supports Baldassarre et al., who also found correlations between middle-distance swimming events and performance in open-water swimming races [28]. In swimming competitions ranging from 50 to 400 m (with durations between 30 s and 4 min), the contribution of both aerobic and anaerobic systems is rather high [29], a phenomenon that is also found in short-distance triathlons. Therefore, athletes who perform better in swimming events with that duration range may have physiological capacities that favour their triathlon race performance. Further studies are needed, however, to confirm this hypothesis.

### 4.2. Correlations between Test Variables and PSL

Regarding the PSL competition variable, it was also observed that high LT1, LT2, and S_L200_ values and low t200 values coincided with low PSL values. The correlation strength between these variables resembled that found for DFT. However, in t200, the correlations were stronger, while in LT1, they were weaker compared to DFT. The reason may be that in events of this nature, it is common to form groups right from the start of the swimming segment [6]. Triathletes with a greater ability to perform short-duration efforts could position themselves in top positions from the beginning of the swim. The rest of the competitors would take advantage of this situation and position themselves immediately behind, reducing their energy expenditure thanks to drafting [9]. Therefore, even if the time differences are not significant, variables such as t200 may be more strongly correlated with PSL as they are closely related to efforts of this nature [30].

Yet, other factors, such as the triathlete’s ability to swim in mass starts, could also influence this relationship. New studies addressing these aspects adopting a multidimensional approach could help improve talent detection procedures through time trial testing [31].

### 4.3. Correlations between Test Variables and Overall Performance

In this sample, no differences were found in LT1, LT2, S_L200_, and t200 among triathletes of different levels, according to Mackay’s criterion [20]. This may indicate that triathlete pool swimming performance is not related to overall performance in the sport. In the same way, differences in pool swimming performance were observed among world-class triathletes with similar competition results [32]. Furthermore, studies like that of Figueiredo et al. have shown that in recent WS races, the overall weight of the swimming leg has a lesser impact on the final results [33].

Nevertheless, a positive association was detected between LT1, LT2, and S_L200_, as well as t200 values and the best overall positions in the WS OD and WS SD races. This would demonstrate that due to the strong correlation between pool swimming performance and triathlon swimming segment performance in WS races, triathletes who swim better are more likely to achieve good results. The reason may be the dynamics of these races. Triathletes with a weaker swimming performance are forced to cycle at higher intensities to catch up with the leaders and this can affect the final result due to greater fatigue accumulation [6]. In a previous study, a correlation was also found between finishing the race in the top 50% and being one of the fastest triathletes in the initial metres (222 m and 496 m) [6]—which could explain the correlation with t200. Consistent with previous research [34], this linear trend was slightly stronger in OD, possibly because of the greater difference in swimming leg splits between the top and weakest swimmers. In N OD, similar relationships can also be observed, although in this case, it may be because in these types of races, small groups of triathletes who succeed at creating significant advantages in the swim tend to form [35]. This prevents the chasing groups from closing the gap during the rest of the race.

No such correlations were found in the other competitive levels measured in this study, possibly because the participants’ lower average performance could result in a less demanding development of the cycling segment, which could favour the chasing of escaped triathletes during this section and lead to lower intensity [36]. This could translate into more similar levels of fatigue at the beginning of the running segment (where the largest differences are found between participants [3]), and therefore, final positions depend less on swimming performance.

### 4.4. Wetsuit Effect

Regarding the results obtained in the RV values in competitions without a wetsuit and with a wetsuit, we can infer from the correlation analysis that in SD triathlons, participants with lower S_L200_ performances benefited more from using a wetsuit. The use of a wetsuit is known to improve swimming speed, submaximal propulsion efficiency, and reduce energy consumption [11], and all these values are strongly related to triathlon swimming leg performance [27]. Also, the use of a wetsuit has been shown to significantly improve biomechanical variables that can influence the speed at which the swimming segment is performed [12]. Therefore, triathletes presenting a weaker performance and efficiency at submaximal velocities may have more room for improvement when using a wetsuit. In elite competitions, wetsuits are allowed when the water temperature is below 20 °C [1]. Therefore, triathletes with low S_l200_ values could prioritise competing in races that take place under these conditions to achieve better results in the swimming leg. Nevertheless, this correlation was not found in OD races, probably because in OD races, swimming shows greater differences between participants and, consequently, predicts the final result more accurately [33]. But more studies are needed to fully understand this phenomenon.

### 4.5. Limitations

Despite these findings, the present study presented several limitations. The races analysed were categorised according to their competitive level to ensure that participants had similar performance levels in all races. However, in some cases, participant level differences in the same category of races may have prevented the discovery of relationships between the studied variables.

The lack of correlation between the variables measured in the test and DFT as well as PSL in the SD WC, SD CC, and OD CC competitions could be explained by factors unrelated to this study. In these competitions, the difference in the participants’ performance level can be significant depending on the race. The competitive levels in CC competitions held in Europe tend to be higher because travel logistics are less demanding, and these races have historically attracted higher-level triathletes. These factors may result in the same participant having lower DFT or PSL values in races outside Europe than in Europe, which could represent a contaminating variable that affects the correlation. In the case of WC races, WC OD races distribute more ranking points than WC SD races [2]. This can lead to the only triathletes participating in WC SD races being those with a greater need to improve their ranking positions or those residing in close proximity to the race. Owing to the latter, the level of these races may vary more than in WC OD races, where the higher number of points may attract higher-level triathletes regardless of race location and timing. Therefore, the same phenomenon found in CC races will occur in WC SD races, but not in WC OD races.

Another significant limitation to consider is the difficulty in standardising the swim segment distance. Indeed, triathletes can become disoriented and swim a distance that is longer than that marked, or the buoys are not placed accurately, which can affect the variables measured in competition, especially TSL. This potential confounding variable may have affected OD races more, as it is twice the distance of SD races, making it more susceptible to measurement error. Finally, although the test closest to the competition was used as a reference, a few weeks sometimes pass between the test and the competition during which performance changes may occur.

## 5. Conclusions

The present study demonstrated that triathlon swimming segment performance can be predicted by a graded multistage pool test using lactate (7 × 200 m every 5 min) as a measure and by conducting a 200 m time trial. Specifically, the strongest associations found were between LT2 and S_L200_ with DFT and PSL. A significant correlation was detected between pool performance parameters and the overall final position in WS OD and WS SD races too. It was equally found that in SD triathlons, participants with lower S_L200_ values obtained greater TSL benefits from the use of wetsuits than participants with higher S_L200_ values.

These findings highlight the relationship between pool and open-water swimming performance, which is of great interest to coaches as they can thus measure the triathlete’s specific capabilities. Furthermore, the fact that swimming with a wetsuit benefits triathletes with lower performance levels could help these athletes to choose races that better suit their characteristics.

However, despite the relationship between pool and open-water performance, coaches should include specific training sessions in their training plans to develop technical skills, especially regarding drafting, self-orientation, and adaptation to constant pace variations.

## Figures and Tables

**Figure 1 jfmk-08-00165-f001:**
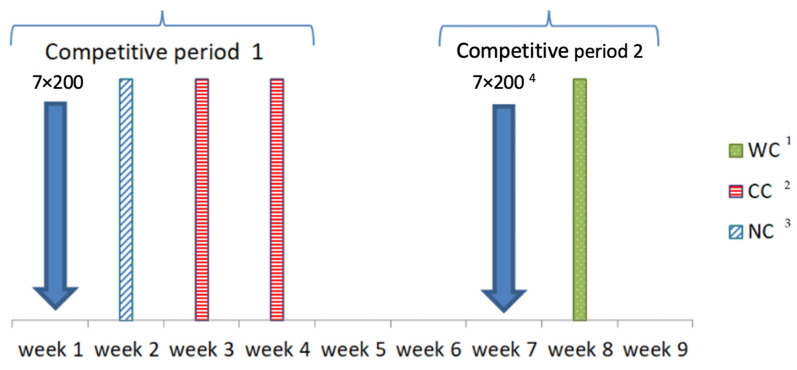
Example of study design for one participant. ^1^ WC: World Cup; ^2^ CC: Continental Cup; ^3^ NC: national championship; ^4^ 7 × 200: 7 × 200 test in pool.

**Table 1 jfmk-08-00165-t001:** Data from the analysed races and the number of participants studied.

Race Level	Total Races Analysed	Average Number of Participants	Total of SD Races	Total of OD Races	No Wetsuit Allowed Races	Wetsuit Allowed Races
Local	5	5.6	3	2	2	3
National	5	3.8	2	3	2	3
Continental Cup	21	2.2	11	10	16	5
World Cup	22	2.1	12	10	15	7
World Series	22	1.7	10	12	17	5

**Table 2 jfmk-08-00165-t002:** Sample data as mean and standard deviation.

Participant Level	Number of Participants	Speed at LT1 ^1^ (m/s) ^4^	Speed at LT2 ^2^ (m/s)	S_L200_ ^3^ (m/s)	200 m Swim Test (s) ^5^
World Class	3	1.31 (0.05)	1.41 (0.05)	1.50 (0.06)	133 (6)
Elite	10	1.31 (0.04)	1.39 (0.04)	1.48 (0.04)	135 (4)
National	5	1.26 (0.02)	1.31 (0.02)	1.38 (0.02)	141 (4)

^1^ LT1: Lactate threshold 1; ^2^ LT2: Lactate threshold 2; ^3^ S_L200:_ Speed in the last test repetition; ^4^ m/s: metres per second; ^5^ s: seconds.

**Table 3 jfmk-08-00165-t003:** Pearson and Spearman correlation coefficient between the test variables and the time difference with the first triathlete after the swim segment across the various race levels.

Race Level	Type of Correlation	Speed at LT1 ^1^ (m/s) ^4^	Speed at LT2 ^2^ (m/s)	S_L200_ ^3^ (m/s)	200 m Swim Test (s) ^5^
SD L ^6^	Pearson	−0.703 **	−0.664 **	−0.716 **	0.713 **
	Spearman	−0.730 **	−0.672 **	−0.745 **	0.601 *
SD N ^7^	Pearson	−0.650	−0.759 *	−0.873 *	−0.947 **
	Spearman	−0.800 *	−0.929 **	−0.829 *	0.857 *
SD CC ^8^	Pearson	−0.546 *	−0.592 *	−0.004	0.260
	Spearman	−0.613 *	−0.608 *	−0.042	0.497
SD WC ^9^	Pearson	−0.407	−0.553	−0.317	0.251
	Spearman	−0.543	−0.626 *	−0.322	0.293
SD WS ^10^	Pearson	−0.710 **	−0.728 **	−0.734 **	0.716 **
	Spearman	−0.663 **	−0.659 **	−0.660 **	0.677 **
OD L ^11^	Pearson	−0.859 **	−0.885 **	−0.892 **	0.873 **
	Spearman	−0.943 **	−0.966 **	−0.914 **	0.747 *
OD N ^12^	Pearson	−0.719 **	−0.761 **	−0.860 **	0.660 *
	Spearman	−0.713 **	−0.710 **	−0.785 **	0.711 **
OD CC ^13^	Pearson	−0.455 *	−0.360	−0.172	−0.220
	Spearman	−0.422 *	−0.334	−0.136	0.243
OD WC ^14^	Pearson	−0.679 *	−0.657 *	−0.802 **	0.711 *
	Spearman	−0.675 *	−0.693 *	−0.892 **	0.735 *
OD WS ^15^	Pearson	−0.692 **	−0.683 **	−0.707 **	0.625 **
	Spearman	−0.595 *	−0.573 *	−0.591 *	0.559 *

*: *p* < 0.05; **: *p* < 0.01; ^1^ LT1: lactate threshold 1; ^2^ LT2: lactate threshold 2; ^3^ S_L200:_ speed in the last test repetition; ^4^ m/s: metres per second; ^5^ s: seconds; ^6^ SD L: sprint distance Local race; ^7^ SD N: sprint distance national race; ^8^ SD CC: sprint distance Continental Cup; ^9^ SD WTC: sprint distance World Cup; ^10^ SD WS: Sprint distance world series; ^11^ OD L: olympic distance local race; ^12^ OD N: Olympic distance national race; ^13^ OD CC: Olympic distance Continental Cup; ^14^ OD WTC: Olympic distance World Cup; ^15^ OD WS: Olympic distance world series.

**Table 4 jfmk-08-00165-t004:** Pearson and Spearman correlation coefficients between the test variables and the position after the swim segment in the different race levels.

Race Level	Type of Correlation	Speed at LT1 ^1^ (m/s) ^4^	Speed at LT2 ^2^ (m/s)	S_L200_ ^3^ (m/s)	200 m Swim Test (s) ^5^
SD L ^6^	Pearson	−0.409	−0.530 *	−0.552 *	0.307
	Spearman	−0.428	−0.551 *	−0.567 *	0.233
SD N ^7^	Pearson	−0.653	−0.767 *	−0.880 **	0.835 *
	Spearman	−0.673	−0.857 *	−0.793 *	0.821 *
SD CC ^8^	Pearson	−0.646 **	−0.811 *	−0.406	0.603 *
	Spearman	−0.692 **	−0.824 **	−0.441 *	0.781 **
SD WC ^9^	Pearson	−0.223	−0.543	−0.499	0.065
	Spearman	−0.282	−0.581 *	−0.424	0.148
SD WS ^10^	Pearson	−0.849 **	−0.831 **	−0.853 **	0.802 **
	Spearman	−0.834 **	−0.708 **	−0.836 *	0.748 **
OD L ^11^	Pearson	−0.905 **	−0.948 **	−0.860 **	0.861 **
	Spearman	−0.905 **	−0.958 **	−0.838 **	0.736 *
OD N ^12^	Pearson	−0.755 **	−0.755 **	−0.880 **	0.710 **
	Spearman	−0.793 **	−0.788 **	−0.852 **	0.797 **
OD CC ^13^	Pearson	−0.454 *	−0.343	−0.262	0.269
	Spearman	−0.440 *	−0.356	−0.258	0.312
OD WC ^14^	Pearson	−0.709 *	−0.607 *	−0.659 *	0.726 *
	Spearman	−0.672 *	−0.576 *	−0.699 *	0.626 *
OD WS ^15^	Pearson	−0.686 **	−0.714 **	−0.728 **	0.651 **
	Spearman	−0.609 **	−0.597 *	−0.632 **	0.612 **

*: *p* < 0.05; **: *p* < 0.01; ^1^ LT1: lactate threshold 1; ^2^ LT2: lactate threshold 2; ^3^ S_L200:_ speed in the last test repetition; ^4^ m/s: metres per second; ^5^ s: seconds; ^6^ SD L: sprint distance local race; ^7^ SD N: sprint distance national race; ^8^ SD CC: sprint distance Continental Cup; ^9^ SD WTC: sprint distance World Cup; ^10^ SD WS: sprint distance world series; ^11^ OD L: Olympic distance local race; ^12^ OD N: Olympic distance national race; ^13^ OD CC: Olympic distance Continental Cup; ^14^ OD WTC: Olympic distance World Cup; ^15^ OD WS: Olympic distance world series.

**Table 5 jfmk-08-00165-t005:** Pearson and Spearman correlation coefficient between the test variables and the final position in the race, across the various race levels.

Race Level	Type of Correlation	Speed at LT1 ^1^ (m/s) ^4^	Speed at LT2 ^2^ (m/s)	S_L200_ ^3^ (m/s)	200 m Swim Test (s) ^5^
SD L ^6^	Pearson	−0.307	−0.381	−0.472	0.289
	Spearman	−0.356	−0.447	−0.540	0.236
SD N ^7^	Pearson	−0.303	−0.392	−0.533	0.527
	Spearman	−0.364	−0.500	−0.577	0.643
SD CC ^8^	Pearson	−0.285	−0.342	−0.052	0.208
	Spearman	−0.269	−0.381	−0.013	0.254
SD WC ^9^	Pearson	0.409	0.133	−0.156	−0.545
	Spearman	0.248	−0.081	−0.049	−0.318
SD WS ^10^	Pearson	−0.505 *	−0.505 *	−0.566 *	0.494 *
	Spearman	−0.573 *	−0.369	−0.600 **	0.494 *
OD L ^11^	Pearson	0.030	0.087	−0.219	0.022
	Spearman	−0.043	−0.097	−0.279	0.030
OD N ^12^	Pearson	−0.575	−0.584 *	−0.709 *	0.596 *
	Spearman	−0.720 **	−0.631 *	−0.782 **	0.648 *
OD CC ^13^	Pearson	−0.125	0.005	−0.132	0.075
	Spearman	−0.031	0.060	−0.064	−0.088
OD WC ^14^	Pearson	−0.014	−0.094	−0.552	0.147
	Spearman	0.009	−0.135	−0.593	0.164
OD WS ^15^	Pearson	−0.611 *	−0.542 *	−0.590 *	0.500 *
	Spearman	−0.611 **	−0.403	−0.623 **	0.396

*: *p* < 0.05; **: *p* < 0.01; ^1^ LT1: lactate threshold 1; ^2^ LT2: lactate threshold 2; ^3^ S_L200:_ speed in the last test repetition; ^4^ m/s: metres per second; ^5^ s: seconds; ^6^ SD L: sprint distance Local race; ^7^ SD N: sprint distance National race; ^8^ SD CC: sprint distance Continental Cup; ^9^ SD WTC: sprint distance World Cup; ^10^ SD WS: sprint distance world series; ^11^ OD L: Olympic distance local race; ^12^ OD N: Olympic distance national race; ^13^ OD CC: Olympic distance Continental Cup; ^14^ OD WTC: Olympic distance World Cup; ^15^ OD WS: Olympic distance world series.

## Data Availability

The data presented in this study are available on request from the corresponding author. The data are not publicly available to preserve the privacy of the performance data of the professional athletes who participated in the study.

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
