# Peer review of "Does Pool Performance of Elite Triathletes Predict Open-Water Performance?"

_jfmk, 2023, doi:10.3390/jfmk8040165_

Round 1

Reviewer 1 Report

Comments and Suggestions for Authors

Authors prepared the article titled “Does Pool Performance Predict Open Water Performance in Elite Triathletes?”. Question in the title is a good approach to interesting potential readers. This type of research is needed for better understanding predicting possibilities from training and tasting conditions to competition performance. But, major revision is required. Main suggestions refers to maximal aerobic speed. It is important, thus it should be resolved before further revision.

Below you see some comments:

Abstract

L12 Add one sentence for the deep background of this study. For example, about triathlon and/or predicting final results from specific tests.

L14 Eighteen male… and three triathletes…

L16 How many swimming tests?

L18 Are you sure that MAS from blood lactate concentration analysis is an appropriate method? Better solution seems to be a 400m swimming test or using ergospirometry analysis. MAs is linked with VO2max.

Introduction

Authors did not show in the introduction any information concerning maximal aerobic speed, physiological meaning, significance of this parameter in training monitoring. It should be filled.

Materials & Methods

Move section subjects before study design.

L146 Section 2.3 Blood lactate concentration data analyses

L154 Cited papers (16,17) did not report information about MAS. Where did you find a method for MAS evaluation from blood lactate? Did you consider other methods to evaluation lactate thresholds eg. presented in the previous article published by Jamnick et al. (2018)? https://pubmed.ncbi.nlm.nih.gov/30059543/

Conclusion

You should use your results from this paper to finally conclude. Show the most important findings not speculations. Sentence starting from “Based on existing literature,…” is good for introduction section or discussion.

Author Response

Thank you for your time reviewing the manuscript. In attachment you can find the file with the responses.

Best regards.

Reviewer 2 Report

Comments and Suggestions for Authors

Well done research but writting should be improved. English is of inappropriate quality, and technically paper is underprepared (comas used for decimal places in text, typewritting mistakes, non-balanced discussion, poorly written Conclusion). Please see comments bellow

Abstract: 

Please start Abstact with study background, not the aim. 

"A correlation index analysis" - Please use some more common term

Please mention that "negative correlation" actually means "positive association"

Introduction

The first paragraf explains the triatlon, which is OK. However, it should be shortened, and more focused. 

Divide second paragraph (line 56 onward) in two; First part explaining the importance of swimming in triatlon, and second one explaining the training procedures. 

Materials and methods

Use "subjects" or "participants", but only one

2.5. Stadistic analysis (???) Please correct

Why did ou use two types of correlation coefficients? Please explain.

Discussion

Please use some subheadings for the Discussion section, otherwise it is hard to follow

In general, Discussion lacks the comparison with previous studies and overview of the literature (not neccesarily researches, books will be suitable also). 

As it is now, discussion is not balanced. Please try to discuss each of your main findings equally. As it is now, it is scattered and not-connected. Try to follow some "red-line" throughout the whole section and not write it separtely, and then merge it.

Conclusions

Conclusion is poorly written. It looks like you got tired. Please rewrite it systematically emphasizing the main findings and practical applications. 

Divide it in 2-23 short balanced paragraphs where you will (2) speak about most important findings and (2) highlight practical application for each. 

Comments on the Quality of English Language

I'm not qualitifed for language quality rreviewing, but you will certainly benefit from proffesional editing of the language. 

Author Response

(The authors gave the same response as above.)

Round 2

Reviewer 1 Report

Comments and Suggestions for Authors

Dear Authors,

Good job, this manuscript has been improved.

Finally, I accept all Authors responses.

Comments on the Quality of English Language

I don't have any comments about quality of English language.

Reviewer 2 Report

Comments and Suggestions for Authors

Nice work. Congratulations!